# Non-Adhesive Liquid Embolic Agents in Extra-Cranial District: State of the Art and Review of the Literature

**DOI:** 10.3390/jcm10214841

**Published:** 2021-10-21

**Authors:** Filippo Piacentino, Federico Fontana, Marco Curti, Edoardo Macchi, Andrea Coppola, Christian Ossola, Andrea Giorgianni, Paolo Marra, Cristina Mosconi, Anna Maria Ierardi, Antonio Basile, Rita Golfieri, Gianpaolo Carrafiello, Giulio Carcano, Massimo Venturini

**Affiliations:** 1Diagnostic and Interventional Radiology Department, Ospedale Di Circolo, ASST dei Sette Laghi, 21100 Varese, Italy; filippo.piacentino@asst-settelaghi.it (F.P.); federico.fontana@uninsubria.it (F.F.); edoardo.macchi@asst-settelaghi.it (E.M.); andrea.coppola@asst-settelaghi.it (A.C.); massimo.venturini@uninsubria.it (M.V.); 2School of Medicine and Surgery, Università degli Studi dell’Insubria, 21100 Varese, Italy; christian.ossola@asst-settelaghi.it (C.O.); giulio.carcano@uninsubria.it (G.C.); 3Neuroradiology Department, Ospedale Di Circolo, ASST dei Sette Laghi, 21100 Varese, Italy; andrea.giorgianni@asst-settelaghi.it; 4Department of Diagnostic Radiology, Giovanni XXIII Hospital, Milano-Bicocca University, 24127 Bergamo, Italy; pmarra@asst-pg23.it; 5Department of Radiology, IRCCS Azienda Ospedaliero—Universitaria di Bologna, 40138 Bologna, Italy; cristina.mosconi@aosp.bo.it (C.M.); rita.golfieri@unibo.it (R.G.); 6Diagnostic and Interventional Radiology Department, Fondazione IRCCS Cà Granda Ospedale Maggiore Policlinico, 20122 Milan, Italy; annamaria.ierardi@policlinico.mi.it (A.M.I.); Gianpaolo.Carrafiello@unimi.it (G.C.); 7Radiology Unit I, Department of Medical Surgical Sciences and Advanced Technologies “GF Ingrassia”—University Hospital “Policlinico-San Marco”, University of Catania, 95123 Catania, Italy; basile.antonello73@gmail.com; 8Department of Radiology and Department of Health Sciences, Fondazione IRCCS Cà Granda Ospedale, 20122 Milano, Italy; 9Department of General, Emergency and Transplants Surgery, Circolo Hospital, ASST Sette Laghi, 21100 Varese, Italy

**Keywords:** artery embolization, artery embolization, EVOH, Onyx, Squid, non-adhesive liquid embolic agent

## Abstract

This review focuses on the use of “new” generation of non-adhesive liquid embolic agents (NALEA). In literature, non-adhesive liquid embolic agents have mainly been used in the cerebral district; however, multiple papers describing the use of NALEA in the extracranial district have been published recently and the aim of this review is to explore and analyze this field of application. There are a few NALEA liquids such as Onyx, Squid, and Phil currently available in the market, and they are used in the following applications: mainly arteriovenous malformations, endoleaks, visceral aneurysm or pseudoaneurysm, presurgical and hypervascular lesions embolization, and a niche of percutaneous approaches. These types of embolizing fluids can be used alone or in combination with other embolizing agents (such as coils or particles) so as to enhance its embolizing effect or improve its possible defects. The primary purpose of this paper is to evaluate the use of NALEAs, predominantly used alone, in elective embolization procedures. We did not attempt a meta-analysis due to the data heterogeneity, high number of case reports, and the lack of a consistent follow-up time period.

## 1. Introduction

Therapeutic endovascular embolization is a well-established treatment both in cranial and extra-cranial districts with a wide variety of embolic devices available on the market. This review focuses on the use of liquid embolic agents and in particular those included in the non-adhesive category. This class includes all those new generation agents based on ethylene vinyl alcohol copolymer (EVOH) composition, Polyvinyl Alcohol (PVA) composition, and Polylactide-co-glycolide (PGC) and Hydroxyethylmethacrylate (HEMA) that have gained popularity due to their “magma-like” flow, enabling good control of the embolic material. Each embolic agent is characterized by its respective strengths and weaknesses and can be used alone or combined with other occlusive agents to enhance its embolic power. It can be said that the perfect embolizing agent does not exist as there is no unambiguous classification or guideline on how to use them.

We can make a first distinction between solid agents, such as coils or vascular plugs (including micro-plugs) that have two major advantages: the first is the ease of use and the second is the detachable control with possibility of repositioning; however, they have the limitation of poor immediate occlusive power. Solid agents also include microparticles, which have the characteristic of being easily conveyed into the target vessel but are poorly controllable and have a discrete risk of migration. 

The second major category is that of liquid embolizers, which include: glue (like acrylates, n-butyl cyanoacrylate), characterized by high and immediate occlusive power but with a high learning curve and greater technical complications [1].

However, they have some disadvantages such as the need to use compatible microcatheters, use of a specific solvent, and the proper costs of the device. In literature, non-adhesive liquid embolic agents have mainly been used in the cerebral district; however, multiple papers describing the use EVOH in extracranial district have been published recently [2,3,4].

The agent that has been used for the longest time is definitely Onyx (ev3 Endovascular, Inc. Plymouth, MN, USA), which appeared in the literature in the 1990s for Arterio-Venous malformation embolization (AVM) of the intracranial district [3,4,5]. Subsequently, it was also used in the abdomen in the early 2010s [6,7,8,9,10]. Another vast field of use in the second half of 2010s was undoubtedly the treatment of Type II endoleaks [11]. 

Squid (Squid-Peri, Emboflu, Gland, Switzerland) is a relatively new non-adhesive liquid embolic agent which has mainly also been used in the cerebral district [12] with some recent application in the abdominal district [13]. PHIL (MicroVention, Tustin, CA, USA) and Easyx (MicroVention, Tustin, CA, USA) are the latest liquid embolic agents to appear on the market for which there is not already solid literature available [14,15]. 

The purpose of this review is to explore the different fields of application of liquid embolic agents in the extracranial district, analyzing through the various published studies what is currently the spectrum of use highlighting the advantages and disadvantages over traditional embolic agents. Use in urgency-emergency was voluntarily excluded to standardize the results collected as much as possible.

## 2. Review Method

The structure of this systematic review and the search methodology employed for this review has been enumerated under the following headings: papers considered for this review had the following predetermined inclusion criteria: (1) Patients undergoing percutaneous embolization for management of different clinical scenario in an extra-neuro district (aneurysm, endoleaks, vascular malformation, etc); (2) Clinical outcomes, follow-up and complications reported; (3) Full-text publications in English available; (4) Publication date between 2010 and 2021. A literature search was performed in July 2021 on PubMed (MEDLINE) for studies which matched the eligibility criteria using the keywords, “artery embolization” or “artery embolization” or “ EVOH” or “ Onyx” or “Squid” or “Phil” or “Easyx” or “non adhesive liquid embolic agent”, every search was conducted then for every chapter of this review. An additional manual search about bibliographies of each included study was done to identify studies not covered by the PubMed search. We did not attempt a meta-analysis due to the data heterogeneity, high number of case reports, and the lack of a consistent follow-up time period.

## 3. Non-Adhesive EVOH Liquid Embolic Agents

This type of embolizing agent appeared in the literature in the early 2000s with preliminary experiences on animal models (pigs) and then came to series on cerebral AVMs and Type II endoleaks (2001 and 2003) [16]. The acronym EVOH stands for ethylene-vinyl alcohol copolymer. Onyx is the precursor of EVOH and is supplied in vials. Each vial contains 1.5 mL of embolizing agent which must be shaken for 20 min before being used (as directed by the manufacturer).

Each vial contains ethylene-vinyl alcohol copolymer (EVOH), dimethyl sulfoxide (DMSO), and tantalum. EVOH is formed of 48 mol/L ethylene and 52 mol/L vinyl alcohol. The polymer is dissolved in DMSO and is available in different concentrations. 

Onyx “6.0%” contains 6.0% copolymer and 94% DMSO, Onyx “6.5%” contains 6.5% copolymer and 93.5% DMSO, and Onyx “8.0%” contains 8.0% copolymer and 92% DMSO. Micronized tantalum powder (35% weight/ volume) is added for radiopacity [17]. The concentration of the copolymer determines the viscosity of the product. The lower the quantity of the copolymer, the less viscous the embolizing agent. When the agent is not very viscous, it has high penetrability in the vessels.

Onyx viscosity of 6.0% and 6.5% is 18 and 20 centipoise (cps; unit of viscosity), respectively, while Onyx 8.0% has a viscosity of 34 cps. The Onyx with concentrations of 18 and 20 has low viscosity and high penetration capacity and is therefore indicated for embolizing small-caliber tortuous vessels or in cases where the catheter cannot reach the target vessel and a more distal penetration is needed. The 34 cp formulation, on the other hand, has high viscosity and poor penetration and is therefore suitable for embolization of large vessels or venous lakes in order to prevent migration [18].

The Squid has similar characteristics to Onyx; however, it has the peculiarity of being produced also with concentration 12 which is a very low-viscosity variant which guarantees great progression. The second peculiarity is that it is produced with a concentration of Tantalum with smaller particles which conditions a more homogeneous solution [12].

A third liquid embolic agent is PHIL (Precipitating Hydrophobic Injectable Liquid), which as a main difference from others, has a iodine component chemically bonded to the copolymer to provide radio-opacity and the nature of the polymer that is not EVOH but PCG and HEMA. Final solidification of the injected agent occurs within 5 min according to the manufacturer. The potential perceived advantages of PHIL are the ready to use formula, avoiding the time-consuming preparation. PHIL, containing no Tantalum, provides fewer artifacts on CT than Onyx [19,20]. 

A further variant available on the market is Easyx, another iodine based embolic agent which presents, as a difference from other agents (Onyx and Squid), the use of Iodine as a radio-opacifying agent [15] and as a polymer is PVA not EVOH.

The main disadvantages of EVOH include the need to use compatible microcatheters, the cost, and not least the mandatory use of dimethyl sulfoxide DMSO. EVOH indeed needs the previous injection of (DMSO) to allow the conduction inside the microcatheter and to guarantee the fluid formula that the agent is still on the target vessel, this solvent can cause vasospasm, endothelial wall damage, and not least pain [21] (Table 1).

## 4. NALEA Alone or Combined with Other Embolic Agents

NALEA embolic agents have the advantage of a high embolizing power and few associated complications (such as catheter entrapment or migration), but they still require a discrete learning curve [11]. EVOH agents can be used alone or in association with each other, depending on the different needs or skills of the user. 

The association between NALEA and particles finds its rationale in the aim to close the central microcirculation (nidus) with the microparticles that must be primarily used and then ensure the occlusion of the feeding arteries with NALEA; this technique is well described in AVM embolizations [21]. 

On the other hand, the association of coils with NALEA is recommended when it is necessary to reduce flow at the level of the target organ to be embolized in order to allow a less extensive use of NALEA with less risk of migration. Moreover, this combination is indicated in cases of visceral aneurysms, as the coils form the skeleton on which the embolic fluid is placed, thus making embolic treatment more solidly effective [22]. A similar level of recommendation is found in the association with vascular plugs. In the latter case, plugs can be used to close the venous outflow in AVMs or in patients with major pelvic varicoceles [23]. 

NALEA, like glue, requires some precautions; indeed, when associated with other embolizing agents, it is always better to use solid agents first, in order to ensure patency and functionality of the microcatheter. NALEA also determines contamination of the microcatheter that could not subsequently be used with other agents, despite a possible washing with DMSO [2].

## 5. Arterio-Venous Malformation

Arterio-venous malformations (AVM) are characterized by an abnormal vascular network with direct communications between the arterial and venous systems; AVMs develop during the early gestational period and usually they grow in teenage and pregnancy [24].

There are also secondary AVMs and the causes are mainly: traumatic outcomes, iatrogenic and post-infection outcomes. This second class differs from the first one for the absence of a real “nidus” and for the presence of a vascular alteration more similar to an arteriovenous fistula, although it can present the same problems and clinical aspects [25].

These vascular lesions could be associated with critical complications such as ulceration, bleeding, cutaneous ischemia, and cardiac failure. 

Actually, there are three different classifications for determining proper treatment of patients affected by AVMs. Schobinger’s clinical classification of AVMs symptomatology has four different classes that can be identified: in the first class, there is no evidence of symptomatology and no recommendation for treatment. If the risks and benefits are correctly evaluated, the treatment is recommended in class II for some specific cases. However, it is recommended for classes III and IV.

The other two classifications are based on the differentiation of AVM angioarchitecture. Both CHO’s three-class and Yakes and Baumgartner’s four-class classification aim to better plan nidus embolization [25,26,27]. For Yakes Type I AVMs, the treatment will be the blocking of the direct fistula between the artery and the vein, better with a mechanical agent such as coils or a vascular plug [27].

For Yakes Type II/III and Cho Type III and IV, the AMVs lesions are characterized by multiple feeding arteries that communicate through a nidus, exiting in multiple draining veins. In these classes, liquid agents are recommended. [27]

Noninvasive imaging is mandatory before treatment. It allows proper confirmation of diagnosis, sets a baseline for the patient, and helps to define treatment. First imaging modality is color Doppler ultrasound, especially for soft-tissue AVMs, which have multiple feeding arteries with increased diastolic flow and an increased systolic-diastolic flow venous return. The color Doppler ultrasound can help target the nidus for direct punctures. MRI is the best examination to evaluate the extension of the malformation in adjacent structures especially for bone involvement. 

Angiography is mandatory before any therapeutic interventions. It allows for evaluating precisely the feeding arteries and draining veins of the malformation and the feasibility of embolization [28].

The main therapeutic options are endovascular or percutaneous embolization and surgery; in some cases, an association of the two is performed. As stated by Soulez et al. [24], a liquid embolics agent could be used to treat AVM, while proximal embolization with coils should be avoided. 

The types of embolic liquid used for AVM embolization are ethanol, glue, and NALEA. Non-adhesive Liquid Embolic Agents are a useful tool in cases of AVM embolization because they harden from the outside to inside; this characteristic allows a deep penetration, known as “magma-like” progression. Non-adhesive Liquid Embolic Agents are nowadays considered the first-line embolic agent for AVM in the central nervous system with higher rates of complete nidus embolization while their use has been extended for peripheral AVM with acceptable safety and effectiveness [29].

Those embolic agents flow toward the target area generating less resistance to injection allowing a much better control [30]. It is crucial that the injection rate is slow and regular because vasospasm could occur if the injection is too vigorous. 

Even though different techniques of NALEA agents injection are described in literature, the rate of injection should be between 0.1 and 0.3 mL/min [30]. 

Gilbert et al. recommends performing the “stop-and-go” technique, with a first 2 min injection in order to create a solid cast around the catheter and then resume the injection to push liquid evoh agent forward [30]. Conversely, in cases where NALEA agents flow in an undesired vascular branch, the same principles should be applied, allowing time for the polymer to solidify and redirect the newly injected onyx to lower resistance areas. As suggested by Yuen Chiu, occlusive balloons or coils can be used to help Onyx penetration while preventing reflux [31].

In 2016, Kilani et al. [32] demonstrated by embolizing a cohort of 19 patients with extra cranial AVMs how the use of non-adhesive embolizing fluids could provide controlled embolization due to slow polymerization enabling deep penetration in the nidus with less risk of catheter gluing due to its non-adhesive nature associated with a high rate of clinical success and low rate of complication.

In the literature, one of the most recent papers with the largest and most heterogeneous case series (14 patients) of extracranial AVM embolization was written by Albuquerque et al. [29], which describes the use of non-adhesive embolizing fluid combined with the use of dual-lumen balloon as an innovative approach associated with excellent clinical results and a low risk of complications, reducing the risk of embolic liquid reflux. A large arteriovenous malformation (AVM) of the pancreatic head with arterial feedings from superior and inferior PDA and anomalous communication with portal vein was embolized using Onyx (3 mL) as described by Grasso et al. in their case report, representing a valid and successful alternative to surgery [33], as reported for AVM causing recurrent pancreatitis and the development of a pancreatic cephalic pseudocyst with hemorrhagic content, successfully treated through embolization of arterial afferents arising from the inferior PDA, gastroduodenal artery, and dorsal pancreatic artery [34].

The latest case report in literature (2019) regards a PDA aneurysm combined embolization with coils detached in superior PDA, a vascular plug for an inferior PDA artery, and an aneurysm sac by Onyx and coils [35].

In addition to the peripheral district, which is perhaps the best represented in the literature, we can certainly talk about the uterine district, which is very well represented in literature; however, if we exclude emergency procedures and vascular malformations, the results refer to a few isolated experiences. In particular, the NALEAs are mentioned in a few papers with a single case series. AVM is a quite rare but potentially cause hemorrhages that can be life-threatening (Figure 1) [36].

Uterine AVMs are most commonly observed after pregnancy that occurs in women with a past history of induced abortion, curettage, uterine surgery, cesarean section, and diethylstilbestrol exposure. However, there is no consensus as to the type of embolization technique and the type of embolic agent. In this regard, several embolic agents have been used including particles, absorbable gelatin sponge, glue, and metallic coil [37]. The first work is that of Barral with a population of 12 patients, using Onyx as an embolizing agent, with excellent results. In this study, it is highlighted how EVOHs are ideal in the tortuosity of the uterine vessels in case of AVM. In addition, two pregnancies are reported after embolization with Onyx [36]. Another work in which some cases of uterine AVM are cited is the review by Venturini et al., in which Squid is used as an embolic agent, with excellent results [13].

## 6. Thoracic Vascular Lesions

Another chapter in the extra-neuro district are embolization procedures in the thorax, mainly involving the bronchial arteries; this district is widely explored in the literature since the first experience in 1973 [38]. However, when the field is narrowed down to non-adhesive embolic agents, the number of articles is significantly reduced.

The first, solid, and focused reported experience is by Bommart in 2012 [39] who performed the embolization using Onyx (ev3 Endovascular, Inc., Plymouth, MN, USA) that is an ethylene-vinyl alcohol copolymer (EVOH)-based liquid embolic agent in 15 patients. This retrospective study shows the good feasibility (except in one artery due to vasospasm related to rapid DMSO injection) of bronchial artery for the treatment of patients with massive bleeding and/or recurrent hemoptysis. Then, it is possible to find only case reports; the last experience using another non-adhesive agent is from Ao et al., using Squid (Squid-Peri, Emboflu, Gland, Switzerland) to get the exclusion of the bronchial arteries [40]. Because hypertrophic abnormal vascularization is usually considered the cause of hemoptysis, authors agree that particles can be used for transcatheter embolic treatment. Coils are a good tool too but in case of a super-tortuous vessel could be difficult to reach the peripheral vascular microcirculation [41]. In case of bronchopulmonary shunt, microparticles or coils may migrate into the pulmonary arteries and in case of bleeding caused by tumor or infectious vascular erosion may not be curative [39].

This new generation of embolizing agents has further advantages, excellent embolizing capacity of the treated vessel, and is able to reach vessels with a diameter of 80 microns (Figure 2). 

Another niche field of application in the thorax is the treatment of pulmonary artery pseudoaneurysm (PSA); there are few papers in literature but only one case report about using Squid combined with coils; Piacentino et al. reported about the embolization of a big pulmonary pseudoaneurysm using soft packing coils to fill the sac; then, they used Squid 18 to occlude the feeding vessel. In this case, coils are used to put in safe the sac and the Squid to avoid a possible recanalisation of the tributary vessel [42].

The last topic found in literature is the embolization of Bronchial artery aneurysm (BAA) that is an uncommon condition and has serious consequences due to possible rupture. BAA rupture is unpredictable and unrelated to the BAA diameter. Izaaryene et al. used a liquid agent (Onyx) because conventional embolization agents such as metallic coils or microparticles would not have allowed safely excluding the most distal BAA and might have carried the risk of recanalization by collateral circulation from systemic non bronchial arteries [8]. Another small but very interesting field of application is the embolization of pulmonary artery for treatment of Hemoptysis caused by lung tumors. As described by Marcelin et al. [43], Onyx could be a valid option to embolize bleeding pulmonary artery erosions and to treat endoleak after pulmonary artery stent graft.

## 7. Endoleak

Endovascular abdominal and thoracic aortic aneurysm repair (EVAR and TEVAR) in patients with a suitable anatomy has become the preferred treatment for abdominal aortic aneurysm [44]. Those endovascular procedures provide a less invasive alternative to open repair with improved outcomes. The most common complication is represented by endoleaks, which may compromise long-term endograft viability, increasing the risk of rupture; therefore, long-term surveillance is needed and in specific cases secondary interventions are requested [45]. Endoleaks are classified on the basis of anatomical origin, among different types: Types I and II are the most common, and are usually treated through an endovascular procedure [46]. Nowadays, a surgical approach is performed only in cases of endovascular that are unsuccessful or a fast sac growth with a high risk of rupture. Among different approaches, endoleak treatment with coil embolization was widely used; however, it is associated with a high rate of endoleak recurrency. For the high reintervention rate, many investigators have refined their technique associating EVOH liquid agents with coils [47]. As shown by Ameli-Renani et al., since 2010, multiple scientific papers describing the use of liquid embolizing agents in treatment of Type I and II endoleaks have been published. In case of Type I (E1), endoleaks represent a vascular emergency due to the high risk of sac rupture. In those cases when the vascular anatomy does not allow an endograft extension or this technique is not successful, embolization with EVOH in association with coils or alone permits to obtain a complete seal of the aneurismatic sac [48]. Type II (E2) endoleaks are found the most following EVAR and also represent the main cause of reintervention [49]. As shown by Chung et al., E2 should be treated when persistent after 12 months from the EVAR procedure and associated with a significant sac size increase [50]. Earlier literature reports the use of coils alone to embolize E2; more recently, multiple articles have reported the use of liquid embolic agents alone or in association with coils. A recent study [47] showed that there is no clinical or statistical difference between the treatment of EL with coils or with Onyx alone, in a population of 17 patients treated with Onyx alone vs. 18 patients treated with coils and glue (There were no significant differences between the two groups regarding age, BMI, or sex). In looking at comorbid conditions, there was no difference in the incidence of hypertension, hyperlipidemia, coronary artery disease, prior myocardial infarction, congestive heart failure, atrial fibrillation, tobacco use, diabetes mellitus, or chronic kidney disease, showing no statistical differences in clinical, efficacy, and safety. In addition, Mozes et al. [44] highlights that EVOH, specifically Onyx, gives similar outcomes to other embolization strategies in the literature and He states that Onyx embolization for management of E2 needs to be judiciously considered, particularly for persisting E2. In addition, Venturini et al., in his review on the use of Squid Peri, described the efficacy and safety in being able to treat Type II endoleak with Squid in four cases, without evidence of complications and with 100% technical success [13] (Figure 3).

Theoretically, to obtain a successful E2 embolization, the endovascular treatment should be aimed close to the nidus cavity, which is easier to achieve with a liquid agent than multiple coils [49]. Furthermore, in specific cases when the vascular anatomy does not allow for reaching the proximity of the nidus, it is possible to inject low viscosity EVOH embolizing agents (Onyx 18, Squid 12) from a proximal location which could flow toward the culprit lesion [51]. As shown by Zener, Marcelin, and Carraffiello, E2 in cases of endovascular failure could be treated with a direct percutaneous sac puncture and embolization with liquid embolic agents [52,53,54]. Among the different NALEA for E2, Onyx is the most widely used followed by Squid. Recently, a case report describing a successful embolization of E2 using Phil has also been published [46,55] with similar results

Worthy of mention is the paper published in June 2021 by Sapoval et al., where eight E2 were treated with a new embolizing agent Easyx with results comparable to the other EVOH agents [15]. In this multicenter study, the clinical success rates were 100% for acute hemorrhage and Type II endoleaks. EASYX (Alameda, CA, USA) is a novel copolymer liquid embolic agent with the particularity of the absence of tantalum that allowed for reduced CT artifacts on imaging follow-up, which was especially useful in patients with Type II endoleaks.

There is a lack of exhaustive literature on the use of NALEA embolizing agents for the treatment of Type Ia and Ib endoleaks; therefore, they will not be addressed in this review. In conclusion, the use of liquid embolizers in the treatment of Type II endoleaks has multiple advantages and some disadvantages. Starting from the latter, NALEA certainly needs a large learning curve to be used; they need compatible microcatheters and have high costs. Furthermore, the mandatory use of the solvent (DMSO) determines not only pain on the injection site but also an unpleasant sweating similar to garlic in the patients treated. On the other hand, the advantage that in some cases makes it indispensable is that of its magma-like nature which allows the target vessel to be embolized even without being able to reach it with the microcatheter, an extremely important condition in the case of non-navigable lumbar circles. Obviously, this last condition is more easily obtainable with low viscosity preparations such as Squid 12 and Onyx 18.

Also with regard to follow-up with CT, current low-density LD preparations allow fewer artifacts. Easyx does not even have tantalum but iodine can be subtracted with modern CT scans.

## 8. Visceral Aneurysms-Pseudoaneurysm

One condition that has been treated with NALEA is splenic aneurysm. A number of papers have recently been published demonstrating the possibility of embolization using NALEA in combination with coils. Both proximal and peripheral splenic aneurysms with different anatomical conformation have been treated in the literature [17,20] (Figure 4).

An interesting series was well reported in literature about nine renal artery aneurysms treated using endovascular techniques over a period of 12 years. Onyx was used as the embolic agent of choice (88.9% cases), with concurrent balloon remodeling. The overall primary technical success rate was 100%; only one case needed a second embolization due to reperfusion (eight years post-first treatment). The Authors concluded that Onyx is an effective and safe primary therapy for trancatheter renal artery aneurysm with high success rate and low morbidity, supplanting surgery as primary therapy [56]. Only case report was found about the treatment of an intrarenal pseudoaneurysm associated with an arterio-venous fistula after Percutaneous nephrolithotomy; Authors described an embolization with Onyx, achieving exclusion of the pseudoaneurysm and AVF with preservation of the remaining vascularization of the kidney [57].

## 9. Tumor Embolization

An important application for these agents used alone or in combination with other devices, showing that satisfactory results are found in vascular lesions’ embolization, such as a giant hepatic hemangioma treated with trans-arterial administration of EVOH and PV [13], a congenital arterioportal fistula treated with a two-step procedure, first with NALEA and then with coils and Amplatzer septal occluder [58] and giant hepatic artery aneurysms treated with EVOH and coils [59], or, in a patient with immunoglobulin G4-related disease, with NALEA, coiling using 20 cm guidewire fragments, and arterial inflow closure with a vascular plug [60].

Two other niche topics, where the use of non-adhesive liquid embolizers has been described, are bronchial artery embolization in pulmonary sequestration and pulmonary artery pseudoaneurysms’ embolization.

Surgical resection has been considered the classic therapeutic approach of Pulmonary Sequestration; however, a life-threatening hemorrhage may occur when the aberrant feeding vessels can not be properly controlled; in these cases, a previous embolization treatment can be useful to control any unexpected bleeding, as described by Venturini et al. using Squid and an Amplatzer plug to minimize the bleeding risk during surgery. Squid was used as an embolic agent with a vascular plug to occlude a large vessel [13]. 

Renal embolization can be performed with various embolic agents, including an absorbable gelatin sponge. There is no consensus on which embolic agent to use, and the final decision depends on each situation and on the interventional radiologist’s preferences [61]. Because their aim is to induce clot formation, the efficacy of absorbable gelatin sponge, coils, and particles depends on the patient’s coagulation status. Liquid agents such as N-butyl cyanoacrylate (NBCA) glue and NALEA copolymer present the considerable advantage of acting independently of any underlying coagulopathy [62]. NBCA glue has a high success rate in renal embolization, but its use is technically demanding and requires a long learning curve. Treatment of acute renal hemorrhage with EVOH copolymer is rarely described in the literature [63]. A relatively large study was conducted by Thulasidasan et al. in 2016 describing renal angiomyolipomas embolization with Onyx [64]. Percutaneous embolization is currently the preferred treatment for enlarging renal angiomyolipoma (AML), and the Authors present mid- to long-term outcomes following
embolization of AMLs with Onyx in seven patients and eleven lesions with circa 6 cm of mean diameter. No hemorrhage from treated lesions was reported, a mean decrease in AML size of 22 mm was seen, and no significant difference between serum creatinine was seen pre- and post-procedure. In some cases, a combined treatment was used, and, in one case, a second round of embolization was performed, although the Authors concluded that Onyx embolization of renal AMLs is effective in the medium to long term. 

A second solid experience was published by Urbano et al. in 2017 [65], who performed in 22 consecutive patients and 25 lesions for symptomatic AMLs or AMLs > 4 cm. Mean AML size in this study was 7 cm. In this study, the peculiarity is that EVOH copolymer was the only embolic agent used. A postembolization syndrome was scheduled in 18.5% of patients, maybe because, when using only Onyx, a large quantity of EVOH was injected. Also for Urbano et al., AML embolization with EVOH copolymer is feasible, safe, and effective (Figure 5).

## 10. Venous Embolization

Non-adhesive embolic fluids are most commonly used in the arterial district, but, in some specific settings, they can also be used in the venous district. The scientific paper with the largest series of vein embolization with non-adhesive embolic fluid was published by Marcelin et al. [66], who demonstrated that the use of Onyx to treat female pelvic varicoceles is an effective and safe approach. Marcelin stated that, in the venous district, these fluids have the great advantage over coils and sclerosing fluids of having good release control, the possibility to see how they distribute on fluoroscopy and both reduced risk of revascularization of the treated venous axis and of dangerous migration toward central venous system. The use of non-adhesive embolic fluids has also been described for the treatment of male varicoceles, with excellent clinical results and improved tolerance to the procedure in the post-embolization period [67,68]. However, it is reported that most patients experienced acute pain during the procedure. For this reason, associated with the high radiation dose administered to the typically young patient, there is no clinical indication for their use in clinical practice [68]. A further use of this type of embolization material has been described by Venturini et al., who demonstrate how squid in combination with coils were effective in the emergency treatment of bleeding esophageal varices in a hepatopathic patient with occluded transjugular intrahepatic shunt (TIPS). In that case report, detachable coils were first placed to provide the scaffold, subsequently compacted by Squid 34, the most viscous formulation, achieving a stable and occlusive plug. EVOH liquid embolic agent can strengthen the embolizing power of the coils. 

Another large field of application in a venous scenario in a non-emergency setting is portal vein embolization (PVE) in order to induce left liver hypertrophy prior to right hepatic resection.

Excluding experimental models [69], first reports have been available since 2018 with Né et al. who described the use of different concentration ethylene vinyl alcohol (EVOH) vials to treat six patients either with a transjugular approach or a percutaneous transhepatic puncture without any complication, enabling the patients to receive a curative surgery [7]. 

A comparison between PVE using EVOH, ethiodized oil, and PVA with surgical vein ligation (PVL) is reported by Biggemann et al. to assess which method induced better regenerative response of remnant liver; they observed 11 patients, even considering that, after percutaneous embolization of target portal branches, they closed the right portal main vein with a vascular plug to prevent embolic agent reflux—using EVOH, liver shows the fastest growth compared to other embolic agents and a significantly higher growth rate compared to PVL [70]. 

Use of EVOH mixed with polyvinyl alcohol (PVA) is reported by Venturini et al. for the treatment of 10 patients (in 1/10 combining embolic agents with coils) obtaining technical success in 10/10 patients (100%), and a clinical success in 8/10 patients (80%) due to insufficient hepatic growth which contraindicated right hepatectomy in two patients [13].

## 11. Extravascular Approach

In some specific cases, when the endovascular approach is not feasible, excellent results have been obtained through the percutaneous approach, as demonstrated by several papers available in literature [71,72,73]. As reported by El Hindy [71], in cases where the lesion’s main blood supply consists mainly of small arteries not feasible for selective catheterisation, or the venous drainage comprises pathologically distended vessels, the CT-guided percutaneous puncture approach of the lesion and application of a non-adhesive embolic fluid may provide a safe alternative to obliterate significant portions of the feeding arteries to the target lesion. The percutaneous approach with non-adhesive fluid has been used with good results by the Clarecon [74] and Lim [75] groups to embolize percutaneously hypervascularized tumours in the pre-surgical step to reduce the high risk of massive intraoperative bleeding. Good results were obtained by the Türkbey [76] and Salaskar [64] groups in the percutaneous embolization of arteriovenous malformations, where the percutaneous approach allows good symptom control and a significant dimensional reduction of the malformation before definitive surgery. 

In 2020, Fanelli et al. [77] reported a case series of forty-one patients with post-EVAR E2 embolized transcutaneously with a 100% success rate. Fanelli states that direct percutaneous puncture of the sac has some advantages over other techniques, as it allows the sac to be completely sealed and, at the same time, the side branches to be embolized. For these reasons, this technique has shown a very good success rate with freedom from re-intervention for recurrent endoleak of 97.9% at 12 months.

Recently, Simoncini et al. [78] published a case report demonstrating the embolization with an echo-guided percutaneous approach of a distal aneurysm of the superior mesenteric artery with a non-adhesive embolic fluid, confirming that, although highly experienced operators are required, this technique can be a valid alternative to the endovascular approach. Obviously, it is necessary to inject the fluid slowly so that it can solidify while it occupies the sac and thus reduce the risks of peripheral embolization. 

Finally, cases have been reported where non-adhesive embolic fluids have been used percutaneously to treat complications of major surgical procedures such as chylothorax [79] post oesophagectomy or biliary leakage [73,80] following liver surgery such as hepatectomy or cholecystectomy. 

## 12. Conclusions

Non-adhesive embolizing liquids compared to adhesive ones have multiple advantages: an excellent embolizing capacity, they could reach vessels with a diameter of 80 microns with a poor tendency of inducing off target embolization and also the risk of microcatheter entrapment is lower. A further advantage, specifically in the pre-surgical setting, is related to the embolization of a lesion in the days preceding the definitive surgical intervention so as to minimize the risk of intraoperative bleeding [81]. On the other hand, the main disadvantages are the need for compatible microcatheters, their higher cost, and, moreover, the mandatory use of DMSO. DMSO is mandatory to keep the embolic fluid dissolved before injection and can cause vasospasm, endothelial wall damage, and pain. Squid and Onyx as embolic agents are widely used in the cerebral interventional field; however, recently, numerous papers regarding their application in the extracranial district have been reported. 

In conclusion, non adhesive liquid embolic agents represent, alone or combined with other embolic devices, a useful and ductile option to treat vascular lesions also in extra cranial districts, adding a range of new therapeutic possibilities that can be “fitted” as a tailored-made suit to each case and patient.

## Figures and Tables

**Figure 1 jcm-10-04841-f001:**
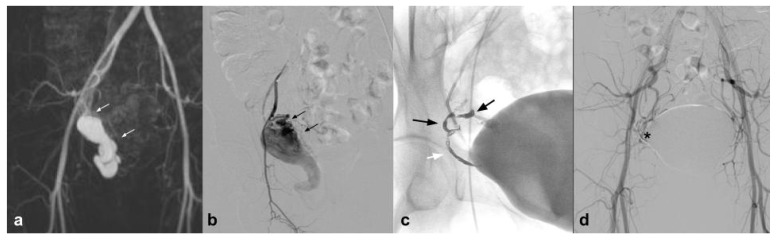
(**a**). Coronal CEMRA MIP demonstrates the presence of voluminous AVM originating from the right uterine artery with aneurysmal venous outflow (white arrows); (**b**) DSA performed with a vertebral shape catheter in the ipsilateral hypogastric artery confirms the presence of the AVM highlighting the “nidus” (black arrows); (**c**) single-shot post-treatment fluoroscopy shows the inferior uterine artery treated (white arrow) with coils (not directly tributary of the AVM) and the presence of a cast of Squid 12 at the level of the middle and inferior uterine artery which were the arterial feeding vessels of the AVM (black arrows); (**d**) post-procedure DSA demonstrates the complete exclusion of the treated AVM (asterisk).

**Figure 2 jcm-10-04841-f002:**
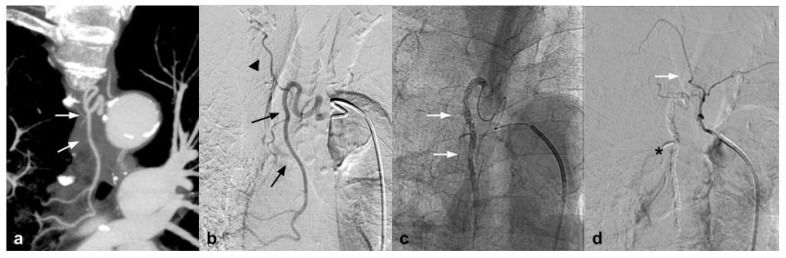
(**a**) Coronal CTA MIP demonstrates hypertrophic bronchial artery with tortuous course originating at 9:00 from the convexity of the aortic arch (white arrows); (**b**) DSA performed with Simmons 1 catheter at the level of the ostium of the bronchial artery confirms the presence of the hypertrophic vessel (black arrow) with a small tributary branch of the right upper lobe (arrow head); (**c**) single-shot fluoroscopy after embolization shows the presence of the microcatheter at the level of the proximal portion of the bronchial artery, with a Squid 12 cast completely occupying the main trunk (white arrows); (**d**) post-procedure DSA control demonstrates complete exclusion of the treated vessel (asterisk) with patency of the small branch for the right pulmonary upper lobe (white arrow).

**Figure 3 jcm-10-04841-f003:**
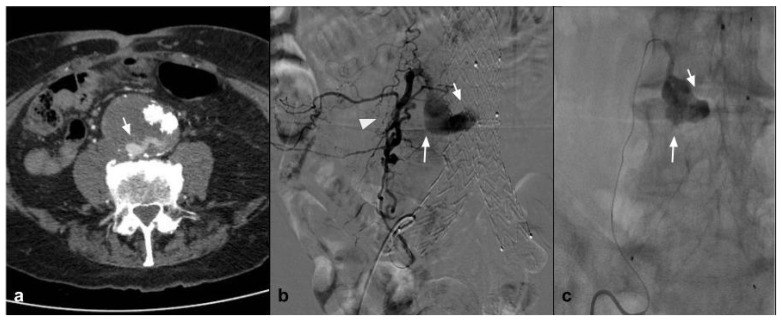
(**a**) Axial CTA shows, the presence of Type II endoleak after EVAR, supplied by lumbar arteries (white arrow). (**b**) DSA performed with microcatheter positioned in a lumbar branch through the ilio-lumbar artery highlights the presence of hypertrophic lumbar circles (white arrow head) with sac refuelling (white arrow). (**c**) Post-procedure DSA shows the cast of Squid 12, which completely occupies the space of the endoleak in the sac (white arrow).

**Figure 4 jcm-10-04841-f004:**
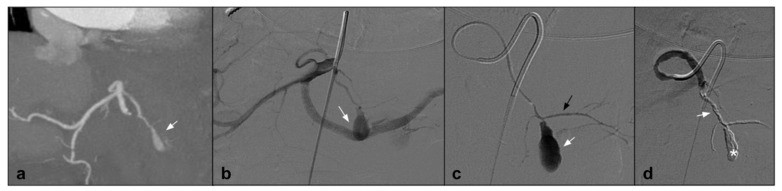
(**a**) Coronal CTA MIP demonstrates post surgical pseudoaneurysm of left gastric artery (white arrows). (**b**) DSA performed with Simmons 1 catheter at the level of the ostium of the celiac trunk confirms the presence of the pseudoaneurysm (white arrow). (**c**) DSA performed with microcatheter in the left gastric artery highlights a saccular dilation (white arrow) of the left gastric artery with regular patency of the efferent vessel (black arrow). (**d**) Post-procedure DSA control demonstrates complete exclusion of the treated PSA (asterisk) with an Onyx 18 cast completely occupying the malacic vessel performing an “endovascular ligature technique” (white arrow).

**Figure 5 jcm-10-04841-f005:**
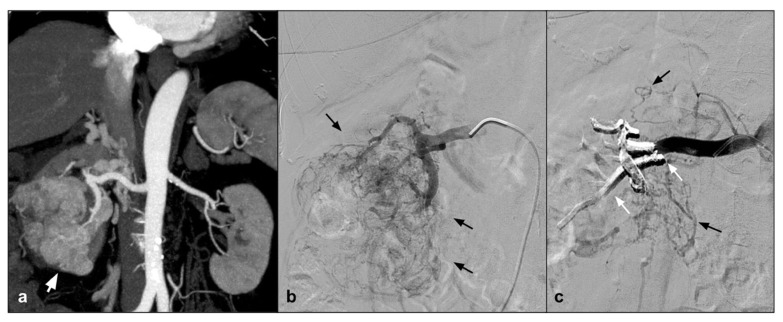
(**a**) Coronal CTA MIP highlights a large angiomyolipoma of the right kidney lower third (white arrow). (**b**) DSA performed with a direct injection in the right renal artery confirms the lesion with an abnormal arterial vascularization (black arrows). (**c**) A post procedure DSA check demonstrates the presence of the Squid 34 cast which completely occludes the tributary arteries of the lesion (white arrows) and preservation of the healthy renal parenchyma (black arrows).

**Table 1 jcm-10-04841-t001:** Nalea embolic agents’ characteristics.

	Onyx	Squid	Phil	Easyx
**Molecule**	Ethylene Vinyl Alcohol Copolymer	Ethylene Vinyl Alcohol Copolymer	Hydroxethylmethacrylate Copolymer	Iodinized Polyvinyl Alcohol Polymer Ether
**Preparation**	20 min shaking	20 min shaking	Ready to use (preloaded in syringe)	Ready to use (vial)
**Radiopacity—Visibility**	Excellent	Excellent	Good	Good
**Embolization power**	Excellent	Excellent	Excellent	Excellent
**Various Formulation (different viscosity)**	Yes	Yes	Yes	No
**Radiopaque agent**	Tantalum	Tantalum	Iodine	Iodine
**Texture**	Chewy	Chewy	Chalky	Chewy
**Injection Time**	Minutes (0.016 mL/min)	Minutes	Minutes	Minutes (0.025 mL/min)
**Ct scan artifacts**	Yes	Yes	No	No
**Solidification process**	Co-polymerisation	Co-polymerisation	Co-polymerisation	Co-polymerisation
**Solvent needed**	DMSO	DMSO	DMSO	DMSO
**Staining (tattoo effect)**	Yes (black colour)	Yes (black colour)	No (opaque-white colour)	No (pearl-white colour)

## Data Availability

No new data were created or analyzed in this study. Data sharing is not applicable to this article.

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
