# Peer review of "Non-Adhesive Liquid Embolic Agents in Extra-Cranial District: State of the Art and Review of the Literature"

_jcm, 2021, doi:10.3390/jcm10214841_

Round 1

Reviewer 1 Report

This is interesting, however some paragraphs needs change and english should be revised.

Please add a table with different liquid agent, name, indication, complication etc..

"10. Percutaneous use":  please change this paragraph and this term; this is not understandable, and paragraph is a mix of procedure.

Figure 4. Coronal and not paracoronal.

Please add an image with vascularization of the kidney after the embolization, because otherwise it looks like you have embolized the whole kidney.

Uterine AVMs:

Please remove uterine AVM from AVM, ther are very rare ! there are mostly 

Retained products of conception.

Figure 1: what is the context ?

SAME: bronchial artery embolization should not be on AVM paragraph !

8. Presurgical embolisation and/or hypervascular lesions embolization:

Please change with embolization with or without surgery.

6. Endoleak.

Add direct puncture of endoleak.

Please add:

Safety and efficacy of embolization using Onyx() of persistent type II endoleaks after abdominal endovascular aneurysm repair. Marcelin C, Le Bras Y, Petitpierre F, Midy D, Ducasse E, Grenier N, Cornelis F.Diagn Interv Imaging. 2017 Jun;98(6):491-497. doi: 10.1016/j.diii.2017.01.003. Epub 2017 Feb 10.PMID: 28196614     

Embolization for persistent type IA endoleaks after chimney endovascular aneurysm repair with Onyx(). Marcelin C, Le Bras Y, Petitpierre F, Midy D, Grenier N, Ducasse E, Cornélis F.Diagn Interv Imaging. 2017 Dec;98(12):849-855. doi: 10.1016/j.diii.2017.04.005. Epub 2017 May 18.PMID: 28528715

7. Visceral Aneurysms-Pseudoaneurysm

Please add images; this is a very interesting indication.

9. Systemic venous district: please change for venous embolization.

Author Response

Reviewer 1
Please add a table with different liquid agent, name, indication, complication etc.
As requested, we have included a summary table with the main characteristics of the non-adhesive
embolic liquids currently available on the market.
"10. Percutaneous use": please change this paragraph and this term; this is not understandable, and
paragraph is a mix of procedure.
the title of the paragraph has been changed to “MISCELLANEOUS : NON ENDOVASCULAR
APPROACH”
Figure 4. Coronal and not paracoronal.
Changed as suggested
Please add an image with vascularization of the kidney after the embolization, because otherwise it
looks like you have embolized the whole kidney.
Unfortunately it is not possible to insert a different post embolization image from the one proposed
because it has not been saved to PACS. However, the embolisation procedure was aimed at
devascularising the lesion in prevision of surgery performed in the same day.
Uterine AVMs:
Please remove uterine AVM from AVM, ther are very rare ! there are mostly
Retained products of conception.
Following your suggestion, we have extrapolated the chapter on uterine AVMs into a dedicated sub-
chapter.
Figure 1: what is the context ?
A 38-year-old female patient complaining of pelvic pain with associated sense of weight who on
transvaginal ultrasound showed a right periuterine "venous lagoon" with demodulated flow on colour-
doppler. She was then referred to our AVM outpatient clinic for in-depth MRI and specialist evaluation.
SAME: bronchial artery embolization should not be on AVM paragraph !
Following your advice, we have excluded thoracic vascular lesions from the chapter on AVMs and
dedicated a separate chapter to them.
8. Presurgical embolisation and/or hypervascular lesions embolization:
Please change with embolization with or without surgery.
Changed as suggested
6. Endoleak.
Add direct puncture of endoleak.

Please add:
Safety and efficacy of embolization using Onyx() of persistent type II endoleaks after abdominal
endovascular aneurysm repair. Marcelin C, Le Bras Y, Petitpierre F, Midy D, Ducasse E, Grenier N,
Cornelis F.Diagn Interv Imaging. 2017 Jun;98(6):491-497. doi: 10.1016/j.diii.2017.01.003. Epub 2017
Feb 10.PMID: 28196614
Embolization for persistent type IA endoleaks after chimney endovascular aneurysm repair with
Onyx(). Marcelin C, Le Bras Y, Petitpierre F, Midy D, Grenier N, Ducasse E, Cornélis F.Diagn Interv
Imaging. 2017 Dec;98(12):849-855. doi: 10.1016/j.diii.2017.04.005. Epub 2017 May 18.PMID:
28528715
Changed as suggested
7. Visceral Aneurysms-Pseudoaneurysm
Please add images; this is a very interesting indication.
Following your advice we have included a case of pseudoaneurysm embolisation of the left gastric
artery embolized with Onyx ( figure 5)
9. Systemic venous district: please change for venous embolization.
Changed as suggested

Reviewer 2 Report

Editing must be reviewed all along (multiple spaces or lacking, et al. not always spelt accordingly)

ref 20 must be cited in the introduction. This article hasn't been read entirely because citation included E2 and bleeding (the ones listed in the summary) but also 3 other indications (PVE, varicoceles and AML) not cited in this review.

Otherwise the review is complete and highly understandable.

Author Response

Reviewer 2

Editing must be reviewed all along (multiple spaces or lacking, et al. not always spelt accordingly)
As requested, we performed a spell and form check.
ref 20 must be cited in the introduction. This article hasn't been read entirely because citation included
E2 and bleeding (the ones listed in the summary) but also 3 other indications (PVE, varicoceles and
AML) not cited in this review. Otherwise the review is complete and highly understandable.
Changed as suggested

Round 2

Reviewer 1 Report

Please, Phil contained polylactide-co-glycolide and polyhydroxyethylmethacrylate) an not EVOH

also easyx contained n iodinized Polyvinyl Alcohol (PVA) and not EVOH

you may write non adhesive liquid embolic agents: NALEA

Figure 5. Please add another picture than the C with opacifiction of the kidney

Please move portal embolization to venous embolization.

9. change for tumor embolization 

move bronchial artery embolization and pulmonary embolization to another paragraph like Thoracic. add: 

2018 Jul;29(7):975-980.

 doi: 10.1016/j.jvir.2018.01.773. Epub 2018 May 5.

Onyx is useful to treat endoleak after pulmonary artery stent graft.

Table 1. add name of the molecule.

Author Response

Please, Phil contained polylactide-co-glycolide and polyhydroxyethylmethacrylate) an not EVOH also easyx contained n iodinized Polyvinyl Alcohol (PVA) and not EVOH.

Correct and appropriate observation, we have changed the technical specifications regarding the two different agents.

You may write non adhesive liquid embolic agents: NALEA

As requested, we have replaced the acronym EVOH with NALEA in the text, as an all-inclusive acronym for the various non-adhesive embolic fluids.

Figure 5. Please add another picture than the C with opacifiction of the kidney
As requested, we sought post-procedure angiography that identified residual renal parenchyma.

Unfortunately, we do not have images with better highlighting of residual non-injury vasculature to the treated kidney. ( Fig. 5 C, EDITED WITH NEW IMAGE).

Please move portal embolization to venous embolization.

Changed as requested

9. change for tumor embolization

Changed as requested

move bronchial artery embolization and pulmonary embolization to another paragraph like Thoracic. add:

Changed as requested

2018 Jul;29(7):975-980.
 doi: 10.1016/j.jvir.2018.01.773. Epub 2018 May 5.
Onyx is useful to treat endoleak after pulmonary artery stent graft.

As suggested, we have expanded on the topic of pulmonary artery embolization by including the bibliographic entry (Marcelin 2018 JVIR).

Table 1. add name of the molecule.

Inserted a row in the table showing the different molecules of the agents.